# Assessment of Trends and Uncertainties in the Atmospheric Boundary Layer Height Estimated Using Radiosounding Observations over Europe

**Fabio Madonna** [1,*] , **Donato Summa** [1,2,*] , **Paolo Di Girolamo** [2] , **Fabrizio Marra** [1] , **Yuanzu Wang** [1,3] and **Marco Rosoldi** [1]

1  Consiglio Nazionale delle Ricerche—Istituto di Metodologie per l'Analisi Ambientale (CNR-IMAA), 85050 Potenza, Italy; fabrizio.marra@imaa.cnr.it (F.M.); yuanzu.wang@imaa.cnr.it (Y.W.); marco.rosoldi@imaa.cnr.it (M.R.)
2  Scuola di Ingegneria, Università degli Studi della Basilicata (UniBas), 85100 Potenza, Italy; paolo.digirolamo@unibas.it
3  School of Earth and Space Sciences, University of Science and Technology of China, Hefei 230026, China
*  Correspondence: fabio.madonna@imaa.cnr.it (F.M.); donato.summa@imaa.cnr.it (D.S.)

**Abstract:** Trends in atmospheric boundary layer height may represent an indication of climate changes. The related modified interaction between the surface and free atmosphere affects both thermodynamics variables and dilution of chemical constituents. Boundary layer is also a major player in various feedback mechanisms of interest for climate models. This paper investigates trends in the nocturnal and convective boundary layer height at mid-latitudes in Europe using radiosounding profiles from the Integrated Global Radiosounding Archive (IGRA). Atmospheric data from the European Centre for Medium-Range Weather Forecasts (ECMWF) ReAnalysis v5 (ERA5) and from the GCOS Reference Upper-Air Network (GRUAN) Lindenberg station are used as intercomparison datasets for the study of structural and parametric uncertainties in the trend analysis. Trends are calculated after the removal of the lag-1 autocorrelation term for each time series. The study confirms the large differences reported in literature between the boundary layer height estimates obtained with the two different algorithms used for IGRA and ERA5 data: ERA5 shows a density distribution with median values of 350 m and 1150 m for the night and the daytime data, respectively, while the corresponding IGRA median values are of 1150 m and 1750 m. An overall good agreement between the estimated trends is found for nighttime data, while daytime ERA5 boundary layer height estimates over Europe are characterized by a lower spatial homogeneity than IGRA. Parametric uncertainties due to missing data in both the time and space domain are also investigated: the former is not exceeding 1.5 m, while the latter are within 10 m during night and 17 m during the day. Recommendations on dataset filtering based on time series completeness are provided. Finally, the comparison between the Lindenberg data as processed at high-resolution by GRUAN and as provided to IGRA at a lower resolution, shows the significant impact of using high-resolution data in the determination of the boundary layer height, with differences from about 200 m to 450 m for both night and day, as well as a large deviation in the estimated trend.

**Keywords:** boundary layer; radiosounding; trends; uncertainties

## 1. Introduction

Atmospheric boundary layer (ABL) processes control energy, water, and pollutant exchanges between the surface and free atmosphere. The ABL is the lowest portion of atmosphere in direct contact with the Earth's surface. The boundary layer top height (BLH) is frequently considered in weather and climate studies, to characterize convective and turbulent processes, cloud entrainment, and air pollutant dispersion and deposition [1,2]. A widely accepted definition given by Stull (1988) defines the ABL as "the part of the

troposphere that is directly influenced by the presence of the Earth's surface and responds to surface forcing with a time scale of about an hour or less". The structure of the ABL can be complex and variable.

The BLH is a key parameter for describing the ABL structure and it is commonly used to characterize the vertical extent of surface-driven mixing, as well as the vertical level at which exchanges with the free troposphere (FT) occur. Moreover, the structure of the ABL is highly variable in space and time, the variability being affected by orography, surface cover, season, daytime and weather. The vertical extent of the ABL, and thus the BLH, may vary from less than a hundred to several thousand meters. The BLH increases with increasing surface temperature and decreasing humidity, which translates into stronger vertical mixing and, typically, lower surface pollution. The growth of the boundary layer is driven by surface sensible heat fluxes, which intensify and dominate over latent heat fluxes, thus leading to increased atmospheric buoyancy. It is to be underlined that, while surface latent heat fluxes moisten the ABL, the primarily contributor to its growth is represented by the sensible heat fluxes [3].

Within the daily evolution of the ABL it is typically possible to distinguish different phases and layers: the daytime convective boundary layer (CBL), the nocturnal boundary layer (NBL), and the residual layer (RL). The CBL is topped by a stable layer above, often characterized by a thermal inversion setting a limit for the mixing with the FT. The CBL is well mixed and adiabatic-conservative quantities like potential temperature, water vapour mixing ratio or aerosol concentration are nearly constant with height [4]. With the progressive attenuation of surface solar heating, near sunset, the ABL enters the so-called nocturnal boundary layer (NBL) phase [5]. This is often characterized by a stable surface layer, about 50–300 m deep. It forms when the radiative heating stops and the radiative cooling stabilize the lowest part of the ABL together with surface friction [6]. At night, turbulence becomes sporadic, mostly driven by wind shear, and the RL can be observed above the NBL [7]. It is worth mentioning that, under particular high or low-pressure conditions, the identification of the ABL height might not be straightforward [8].

Estimates of the BLH are crucial for air quality management and forecasting. In this context, the vertical extent of surface-driven mixing is of particular interest, so that the CBL is often called a mixed or mixing layer by the modeling community. The term "mixing" must be preferred with respect to "mixed" as the use of mixed should be reserved for layers where the mixing process has led to constant value of potential temperature and other scalars like water vapour mixing ratio.

In the numerical models, the parameterization of the mixing layer (and entrainment zone) is diagnostic and typically based on a parcel method or on the bulk Richardson method, Ri [9–11], applied on the modeled vertical profiles of meteorological parameters. When estimated from observations, instead, the BLH can be estimated using several measurement techniques, such as radiosondes, wind profilers, light detection and ranging equipment (lidars) and ceilometers as well from other Earth Observations (EO) techniques, mainly Global Navigation Satellite System-Radio Occultation (GNSS-RO) and infrared sensors. The applied methodologies are several and mainly based upon Ri, the parcel method or on the identification of gradients in vertical profiles of an atmospheric parameter [9].

BLH estimations are provided also by atmospheric reanalysis products, available from the climate services. Although atmospheric reanalyses have proved to be valuable for the study of climate when used appropriately, their accuracy can considerably vary depending on the location, time period, and variable considered [12]. Even less evidence and fewer comparisons exist for the ABL [13] and in particular for the study of BLH trends.

The study of ABL is typically focused on different time scales and cycles (diurnal, seasonal, annual, etc.), depending on the different application. Climate applications require long-term stability of the ABL estimates to properly interpret the specific feedback mechanisms linking basic climate variables, such as surface temperature and the low-level clouds [14].

Radiosondes represent the most widely used observational dataset for the study of the ABL at global and regional scales. A large number of algorithms have been developed to infer the ABL height, mainly distinguishable based on the detection of a specific vertical gradient or the exceedance of a critical threshold value in the turbulence process (e.g., Ri). The comparison of these algorithms revealed how climatological BLH estimates are subject to both parametric and structural uncertainties associated with methodological aspects, mainly the choice of the estimation method, the vertical resolution of radiosounding data, and the inclusion or exclusion of surface-level observations. Each of these contributions may generate uncertainties, systematic rather than random, of the order several hundred meters [9,11]. In addition, as some components of the climate system respond slowly to change, the climate system naturally contains persistence which may affect also the BLH trend estimates. This becomes clear when there is a certain degree of autocorrelation in the data which must be taken into account to avoid an overestimation of trends [15].

In this paper, trends in the BLH estimates from radiosoundings in Europe covering the mid-latitude domain are studied for the period 1978–2018 using the method of the vertical gradient of potential temperature. The selection of the geographical domain is largely driven by the intent to restrict the focus to a single climate regime over a region with a high density of radiosounding stations with long historical data records available. The effect of parametric, i.e., sampling, uncertainties due to the vertical resolution of the radiosounding profiles and time series completeness is discussed.

Observational trends are also compared with trends estimated from the Atmospheric data from the fifth generation European Centre for Medium-Range Weather Forecasts (ECMWF) ReAnalysis v5 (ERA5), which are based on the bulk Ri method. Despite the fact that the BLH retrieval method adopted by ERA5 differs from the potential temperature gradient method considered in the Integrated Global Radiosounding Archive (IGRA) data, the comparison is valuable because the latter is widely used in literature for radiosoundings data records and has been recommended for comparisons with climate models [9]. As a consequence, the assessment of the deviations between ERA5 and IGRA is relevant for both climate studies and modeling evaluation, especially when long historical data records are considered and the unavailability or discontinuity in near-surface wind data or the large instrumental uncertainties may increase the structural uncertainty or require assumptions to be made.

After a description of the datasets in Section 2, Section 3 introduces the methodologies adopted for the estimation of trends in the ABL. Section 4 discusses the comparison with ERA5 and the effects of parametric uncertainties. Finally, conclusions and outlooks are provided.

## 2. Datasets

### 2.1. Integrated Global Radiosounding Archive (IGRA)

Global radiosoundings are obtained from IGRA, which is the most comprehensive, authoritative collection of historical and near-real-time radiosonde and pilot balloon observations from around the globe, maintained and distributed by the National Oceanic and Atmospheric Administration's National Centers for Environmental Information (NCEI). Data are taken from IGRA data version V2 released in August 2016 [16,17], which has several quality improvements with respect to the previous version (V1), such as an increased spatial coverage, and incorporates data from a considerably larger number of data sources, with an increased data volume by 30%.

The BLH is estimated from the IGRA radiosoundings temperature profiles considering the location of the maximum potential temperature vertical gradient [18], scanning the profile from the ground level upward. The IGRA profiles are provided at 16 standard pressure levels plus a variable number of significant levels [19]. The potential temperature of a fluid parcel at pressure p is the temperature that the parcel would attain if adiabatically brought to a standard reference pressure value p0, usually 1000 hPa (https://rda.ucar.edu/datasets/ds627.1/docs/Pressure_and_isentropic_levels/, accessed on 22 February 2021).

The potential temperature is denoted with θ(z), and for a gas well-approximated as ideal, is given by:

$$\theta(z) = T(z)\left(\frac{P_0}{P(z)}\right)^{\frac{R}{C_P}} \tag{1}$$

where T(z) is the current absolute temperature (in K) of the parcel at altitude z, R is the gas constant of air, and $C_P$ is the specific heat capacity at a constant pressure. $R/C_P$ is equal at 0.286 for air. The level of the maximum vertical gradient of θ(z) [5,6] is indicative of a transition from a convectively less stable region below to a more stable region above.

For the purpose of studying BLH trends, nighttime and daytime data have been evaluated separately in order to assess climate effects in both the CBL and the NBL. Hereafter, the concept of data completeness is assumed as the fraction of available measurements to the number of expected measurements. This study considers only BLH determination at 00 UTC and 12 UTC, and therefore it is assumed that the number of expected radiosounding measurements per day is one. Figure 1 shows the IGRA data in the mid-latitude portion of the European continent filtered to select only those stations offering time series with a data completeness larger than 75% over the period 1978–2018. The list of the stations available in the same spatial domain, along with the corresponding latitude, longitude, operation period and fraction of total launches (night and day) is reported in Appendix A. Afterwards, trends are estimated using a robust non–least square method for those stations only with a minimum of 15 radiosounding profiles available per month for at least a decade in the period of study.

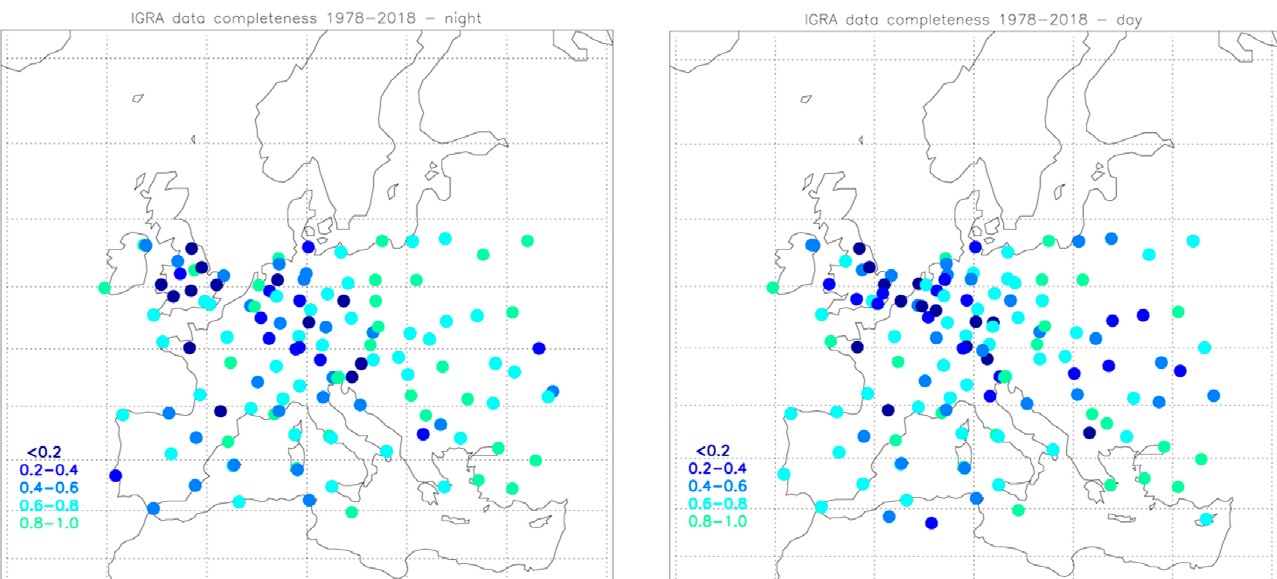

**Figure 1.** Map of the radiosoundings stations considered in the study. Colors represent the degree of completeness for the nighttime (**left** panel) and daytime (**right** panel) time series available from the European mid-latitude stations. Data completeness is calculated as the fraction of the night and day measurements available since 1 January 1978.

Cases with precipitation or low clouds have not been excluded from the dataset and their effect is considered as a contribution to the seasonal variability at each site. Across the dataset, BLH height was observed to never exceed 2900 m above the ground.

The selected domain, covering a time zone from GMT+0 to GMT+2, ensures that daytime radiosounding data allow monitoring the CBL when its vertical extent is maximum or close to its daily maximum, and well-mixed, thus minimizing the effect of the time zone on the development of the ABL and, as a consequence, on the homogeneity of trend estimations.

*2.2. ERA5*

The BLH estimates obtained from the radiosounding profiles have been also compared with those provided in the ERA5 atmospheric reanalysis. ERA5 is the latest climate reanalysis produced by ECMWF, providing hourly data on regular latitude–longitude grids at $0.25° \times 0.25°$ resolution [20], with atmospheric parameters on 37 pressure levels. ERA5 is publicly available through the Copernicus Climate Data Store (CDS, https://cds.climate.copernicus.eu, accessed on 15 January 2021).

To carry out the comparison, the nearest ERA5 grid-point has been considered for each radiosounding station, assuming that the representativeness uncertainty associated with the use of the nearest grid-point to be lower or comparable to that affecting other comparison methods (e.g., kriging, bilinear interpolation, etc.). The mixed layer parametrization makes use of a boundary layer height inferred from an entraining parcel model. In order to obtain a continuous field, also in neutral and stable conditions, a bulk Richardson number (Ri) method is used as a diagnostic, independently of the turbulence parametrization [9]. Under these assumptions, the BLH is defined as the lowest level at which the bulk Ri, reaches the critical value of 0.25, which should be suitable for both convective and stable boundary layers. The bulk Ri is defined as the ratio the consumption of turbulence divided by the shear production (i.e., turbulence kinetic energy caused by wind shear) of turbulence (usually negative). The flow is assumed to be turbulent for negative values of the bulk Ri, while it is assumed to be laminar if the bulk Ri exceeds the critical value. The ABL height is determined through a vertical scan from the surface upwards. A full description of the algorithm can be found in the ECMWF confluence website (https://confluence.ecmwf.int, accessed on 22 February 2021) Recently, Zhang et al. (2014) found that the optimal critical values of the bulk Ri increase with increasing ABL instability and the selected critical value may affect the accuracy of the BLH estimate. This structural uncertainty is considered in this study as one of the possible uncertainty sources, but it is not quantitatively assessed.

*2.3. Caveats*

The application of the bulk Ri method to the radiosounding data requires the assumption of several approximations in the original algorithm, which can lead to erratic and unreliable results [3]. For example, the lack of information to parameterize surface roughness at each measurement site does not allow estimation of friction velocity and, therefore, surface frictional effects are neglected in the computation of the bulk shear [3]. In addition, the historical radiosonde observations available from IGRA do not include near-surface winds: this implies the assumption of a null value for the near-surface wind velocity. Considering these limitations, it is advisable to estimate the BLH from radiosounding observations using alternative methods, able to use reduce uncertainties due to missing data or to strong assumptions. Methods based on the identification of vertical gradients in specific atmospheric variables are the most appropriate choice.

## 3. Methodology

Estimation of trends are obtained using a linear regression estimator, which can be parametric or not. Generalized Least-Square (GLS) [21] and robust methods like Theil-Sen [22] estimators are the most common choice adopted in literature. In this study, the calculation of ABL trends is carried out after a preliminary evaluation of time series autocorrelation [15,23], which is often present in climate time series (e.g., temperature).

We assume the mathematical representation of a climate variable in terms of its mean and variability terms through the following additive model:

$$X(i) \; = \; X_{tr}(i) \; + \; S(i) \; + \; \varepsilon(i) \tag{2}$$

where the variable $X(i)$, is, hence, described by the time-dependent component $X_{tr}(i)$, the trend, and the seasonal component $S(i)$. The residuals $\varepsilon(i)$ has a distribution with null mean and standard deviation $\varepsilon(i)$, but the full shape of the distribution is not prescribed.

To extend the full statistical description of the climate system to one that includes not only the first statistical moment (expectation) or the second (standard deviation), but also higher orders and extremes, other more sophisticated models may be considered [24].

From Equation (2), the seasonal cycles can be removed by subtracting monthly anomalies and thus obtaining the following equations for the monthly anomalies of $X(i)$:

$$X'(i) = X_{tr}(i) + \varepsilon(i) \tag{3}$$

Climate variables often show residual distributions not consistent with a Gaussian distribution. This is also due to the fact the residuals often exhibit autocorrelation, i.e., if $\varepsilon(i)$ is positive, then $\varepsilon(i+1)$ is likely also positive. This is defined as a "memorizing ability of climate" acting on many timescales [24]. This is also called persistence or serial dependence. The autocorrelation in the residual must be investigated and taken into account in the estimation of trends and the related uncertainty. If neglected, there is a risk to overestimate trends and underestimate the uncertainties.

Even when autocorrelation is present, the linear regression coefficients are unbiased, but they are not necessarily the estimates of the population coefficients that have the smallest variance. In this paper, the autocorrelation is taken into account by assuming a first-order autoregressive model AR (1) [15,23,24]. Least absolute deviation (LAD) is used to estimate decadal trends. LAD is a resistant and non-parametric regression method fitting the paired data to the linear model using a robust and resistant method. The technique is based on an algorithm by Barrodale and Roberts (1974) [25]. Comparisons with the classical least squares regression demonstrated that the proposed algorithm is more resistant to the existence of outliers and gives more intuitive results with less sensitivity to outliers [26,27].

If we suppose that LAD assumptions hold, except that there is autocorrelation, i.e., when the expected value $E[\varepsilon(i), \varepsilon(i+h)] \neq 0$, where $h \neq 0$, for a first-order autocorrelation process $\varepsilon(i)$ can be expressed as [28]:

$$\varepsilon(i) = \rho\varepsilon(i-1) + \delta_i \tag{4}$$

where $\rho$ is the first-order autocorrelation coefficient, i.e., the correlation coefficient between $\varepsilon(1), \varepsilon(2), \ldots, \varepsilon(n-1)$ and $\varepsilon(2), \varepsilon(3), \ldots, \varepsilon(n)$, and $\delta_i$ is an error term with namely $E(\delta_i) = 0$, the variance $\text{var}(\delta_i) = \sigma_\delta^2$ and the covariance $\text{cov}(\delta_i, \delta_j) = 0$ for all $i \neq j$.

The trends are then estimated using the LAD fitting method in the equation:

$$X_i - \rho X_{i-1} = A + B(t_i - t_{i-1}) \tag{5}$$

where $X$ is the modelled variable, $t$ is time, $A$ and $B$ are the coefficients of the linear regression. Note that since $\rho$ is a correlation coefficient, it follows that $-1 \leq \rho \leq 1$ and the noise process $\delta_t$ is stationary. This model allows the residuals to be autocorrelated among successive observations [23,24].

## 4. Results

### 4.1. Autocorrelation and Residuals

Before estimating the decadal trends obtained from IGRA and ERA5 data, we preliminarily studied the autocorrelation of both the time series and the residuals (Equation (2)). In the top panel of Figure 2, the autocorrelation of the IGRA time series for one of the selected stations is illustrated (Wien/Hohe Warte, Austria). The plot shows the presence of peaks due to the increase of correlation associated with the ABL annual cycle, along with an additional signal with a longer periodicity. In the middle plot of Figure 2, the autocorrelation of residuals $\varepsilon(i)$ (Equation (4)), calculated on the monthly anomalies, are shown. Although small, the autocorrelation values are still statistically significant and, therefore, must be removed before the trend estimation using Equation (4). Residuals are still affected by the other periodic signal components. The bottom panel of Figure 2 shows the autocorrelation of residuals after the removal of the lag-1 autocorrelation, which

eliminated the influence of the autocorrelation terms in the BLH trend estimations. The removal of the lag-1 correlation term must be pursued with care to avoid over smoothing of the linear regression and the consequent removal of climatic signals. In the presented trend analyses, the lag-1 autocorrelation term has been removed using the approach discussed in Section 3 for all the time series of the BLH monthly anomalies when it exceeds a value of 0.2.

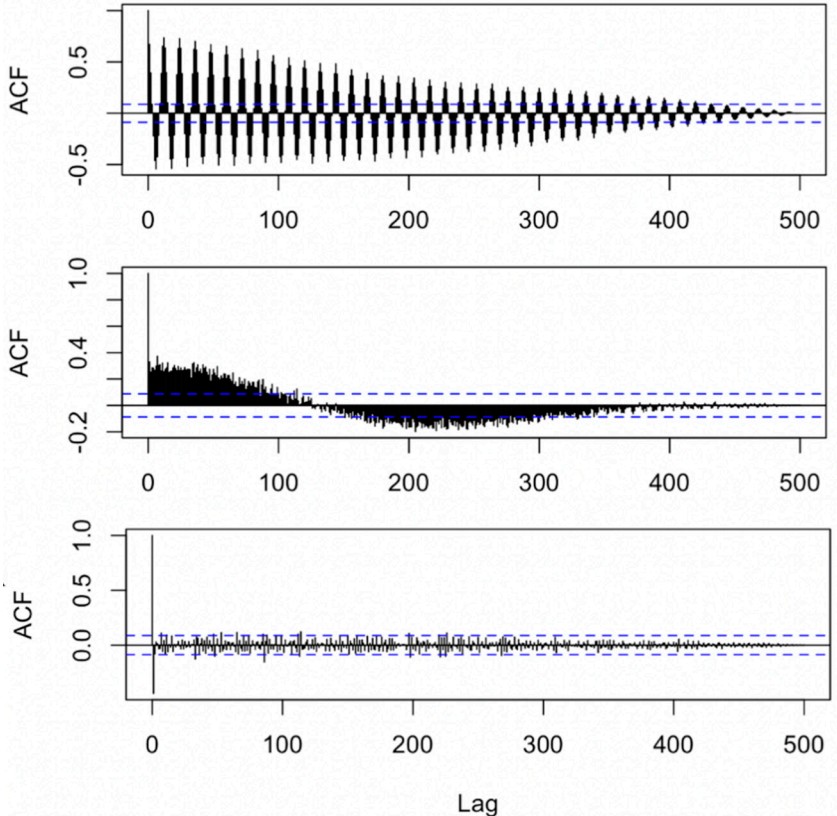

**Figure 2. Top** panel, autocorrelation of the boundary layer top height (BLH) monthly averages for the Wien/Hohe Warte station in Austria (WMO index = 11,035, 48.2486° N, 16.3564° E, 200 m asl) since 1978; **middle** panel, autocorrelation of residuals of BLH monthly anomalies linearly fitted with the least absolute deviation (LAD) regression method; **bottom** panel, same as **middle** panel after the removal of the lag-1 autocorrelation term.

*4.2. Probability Density Functions*

In Figure 3, the density functions of the nighttime and daytime monthly averages of the BLH retrieved from IGRA and ERA5, for all the IGRA selected stations and the whole period of study in Europe, are shown. The density functions confirm the large differences between the two estimation algorithms, based on the identification of vertical gradients in potential temperature and the exceedance of the critical threshold value in bulk Ri, respectively. ERA5 data show a right-skewed distribution, at both night and day, with median values of 350 m and 1150 m respectively, while IGRA density function is symmetrical and with similar shape at both night (NBL) and day (CBL), with median values of 1150 m and 1750 m, respectively. The larger the median values of the CBL height compared to the NBL are mainly due to the solar heating [5]. The differences between the ERA5 and IGRA density functions are larger at night, with IGRA NBL height always higher than ERA5, while during the day the density functions have a more similar shape and the difference between the two functions appears quasi-systematic. Discrepancies between the median values are consistent with results discussed by Seidel at al. (2010). Lower ABL heights are typically associated with the bulk Ri method rather than with the potential temperature gradient method.

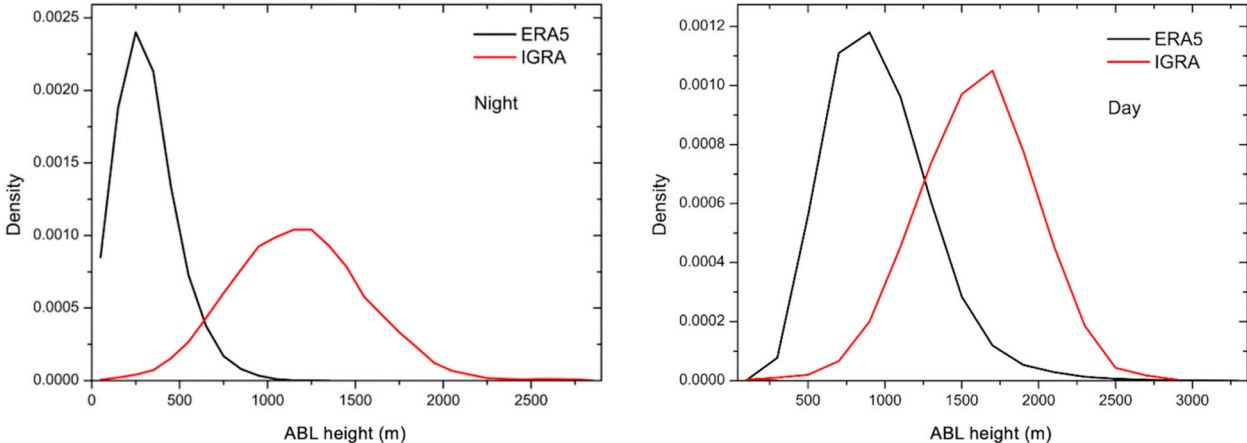

**Figure 3.** Density functions of the BLH monthly averages for the night (**left** panel) and day (**right** panel) time data of ERA5 (black lines) and Integrated Global Radiosounding Archive (IGRA, red lines). Different vertical scales are used to permit the integral plotting of the pair of density functions.

### 4.3. Decadal Trends

The calculation of decadal trends based on IGRA and ERA5 data, at the European stations selected in Section 2, are shown in Figure 4. The comparison of the decadal trends reveals an overall similar behavior for the two datasets at night, while some differences are present during the day. Nighttime decadal trends for IGRA have not a prevalent sign and oscillate between −10 and 20 m per decade, while for ERA5 values range within −10 and 10 m per decade for ERA5 and mainly positive (for both the dataset with the exception of few sites only). The trends for IGRA also show that positive values are more frequent at coastal stations while negative for the continental. During the day, the trend values of CBL height are mainly positive for ERA5, while IGRA has a narrow prevalence of positive values. It can be noted that ERA5 values are also less homogeneous than IGRA with values ranging between −30 and 50 m per decade, while IGRA values ranges between −10 and 10 m per decade, with the exception again of two sites with values smaller than −10 m. The larger oscillation in the ERA5 data can be due to different reasons: structural uncertainties of the bulk Ri method due to the estimation of the friction velocity [9] and the applied critical value [11] or parametric uncertainties linked to the vertical resolution of the ERA5 profiles, lower than for IGRA. For both IGRA and ERA5, specific spatial patterns in both the NBL and CBL trends cannot be distinguished.

Results from the use of the bulk Ri method to estimate the CBL from IGRA data has been already discussed in literature [29] for a limited number of stations at the global scale. The considered data sample includes a number of European stations where the decadal trends inferred from the IGRA data are prevalently positive although some sites have trends with negative values, in general agreement with the results presented above. This might indicate that the CBL height differences shown in this work between the two different methods applied to IGRA and ERA5 are substantially due to parametric uncertainties or to the choice of the bulk Ri critical values. Such uncertainties have already been examined in Seidel et al., 2010, but not for trend estimations. Structural uncertainties, which contribute the difference shown in Figure 3, were also qualified by Seidel et al. (2010) as more systematic than random and, therefore, very relevant when comparing radiosonde-based climatological estimates with those from models.

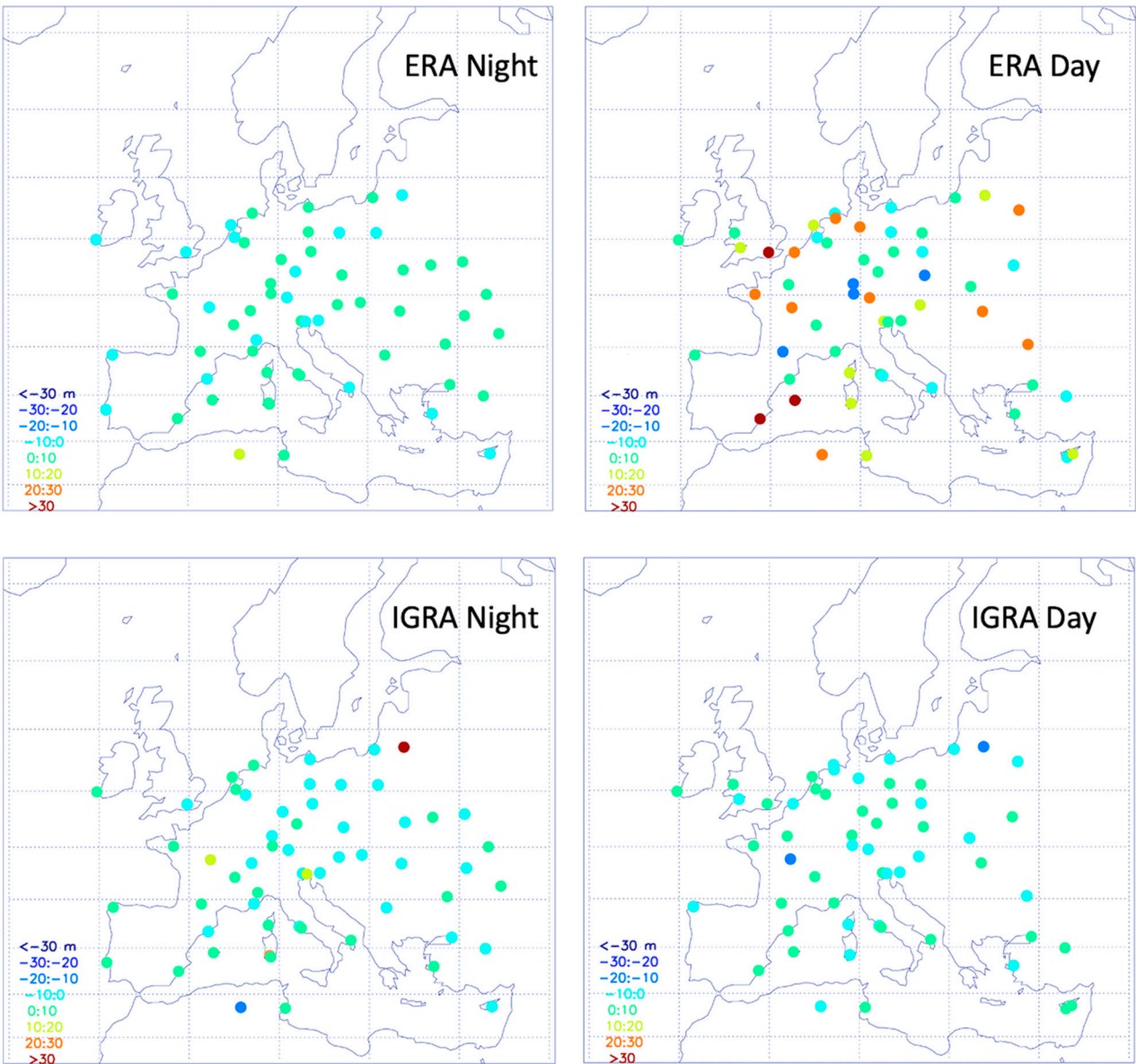

**Figure 4.** Maps of decadal trends of the nocturnal boundary layer (NBL, **left** panels) and convective boundary layer (CBL) height (**right** panels) retrieved from ERA5 (**upper** panels) and IGRA (**lower** panels) in the period 1978–2018. Color legend is reported in the bottom left corner of each panel. Dots are representative of the location of the European Union (EU) mid-latitudes radiosounding stations with a data completeness larger then 75%.

To quantitatively investigate the consistency between the IGRA and ERA5 BLH trends in the selected domain, in Figure 5 the time series of BLH mean monthly anomalies for all stations in Figure 4 are shown with the corresponding trend estimation. Spatial autocorrelation has not been investigated and its contribution has been assumed isotropic in the investigated spatial domain. Figure 5 shows that the NBL height trend for IGRA is of −2.7 m per decade, while for ERA5 is almost neutral and equal to 0.2 m per decade. For the CBL, the IGRA trend is −4 m per decade, while for ERA5 it is about 5 m per decade. In both plots in Figure 5 the median absolute deviation (MAD) is also included which quantifies the variability range of linear regression residuals. The daytime comparison also shows a broader variability in the ERA5 mean monthly anomalies, in agreement with the variability of the trends shown in Figure 4. Moreover, both day and night anomalies show a larger variability in 80 s and 90 s mainly because of the lower resolution and availability of

both the IGRA data and the input data to the ERA5 data assimilation system, respectively. In the most recent decade, instead, the BLH height anomalies show a good agreement.

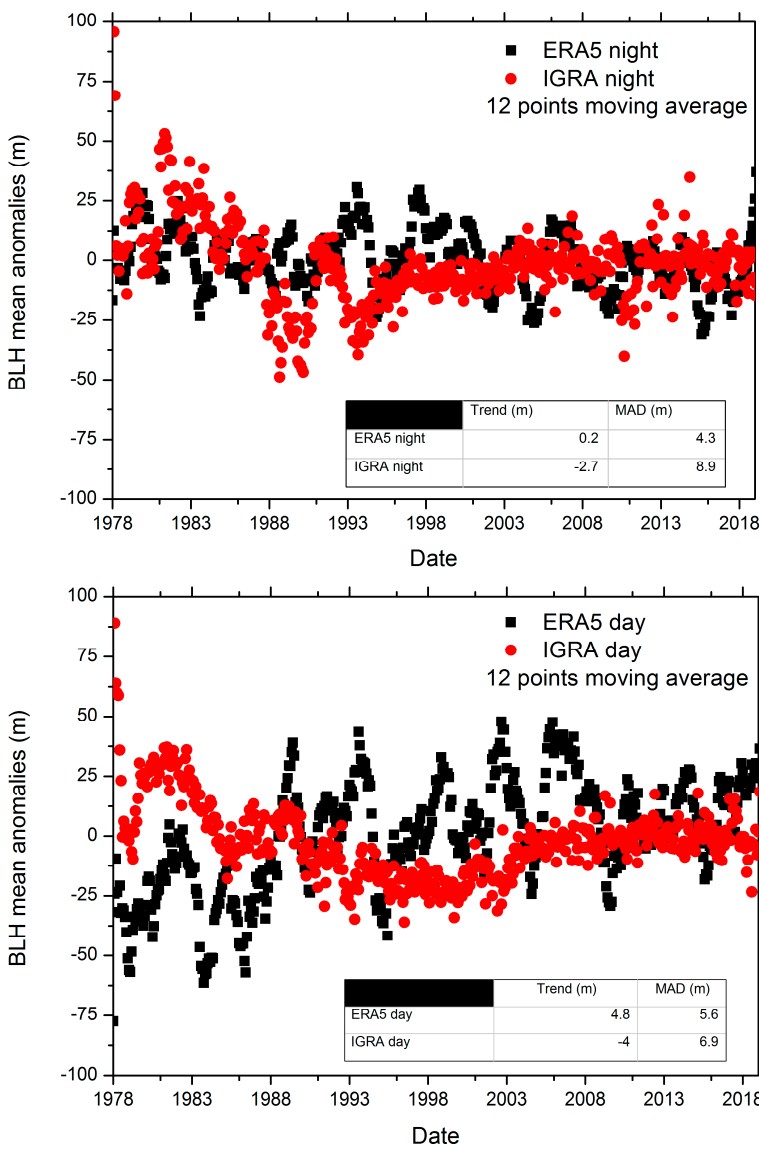

**Figure 5. Top** panel, comparison of the ERA5 (black dots) and IGRA (red dots) nighttime BLH mean monthly anomalies for the stations reported in Figure 4; **bottom** panel, same as **top** panel for daytime anomalies. Both panels embed a table with the estimation of the decadal trend and the median absolute deviation (MAD). All the shown anomalies are obtained applying a 12-point moving average on the original time series.

It is intuitive to relate the trends of the BLH and the well-known increase of the Earth's surface temperature observed in recent decades over most of the globe [30]. The increase of surface temperature over the time period considered in the present study (1978–2018) for Europe is about 2.0 K [31,32]; https://www.eea.europa.eu/data-and-maps/indicators/global-and-european-temperature-10/assessment, accessed on 22 February 2021): this increase has also contributed to the trends of the BLH, although this is one of the forcing components only. In this regard, it is to be considered that coupling mechanisms and feedbacks between soil moisture and surface fluxes play a major role in ultimately determining the ABL properties and its vertical development [33], with variable effects in different geographical areas. Climate models forecast an increasing role of soil moisture in the development of thermally driven boundary layers in a warming climate [34]. This mechanism can also support ex-

plaining the projected climate model trends toward an increase of frequency and intensity of hydrogeological extreme events [35].

### 4.4. Effect of Data Completeness and Sampling on the Trend Uncertainty

Measurement density and distribution in a long time series, with the presence of gaps (i.e., missing data), may have a critical role in the uncertainty of estimated trends in dependence on the time and spatial variability of the investigated atmospheric variable. In Figure 6, it is shown the comparison between the BLH mean anomaly time series considering all the stations providing a data completeness in the period 1978–2018 of 50%, 75% and 90%, respectively, for both day and nighttime measurements. In addition, the corresponding median trend and the MAD for the selected stations (68, 56, 46 points respectively) are reported.

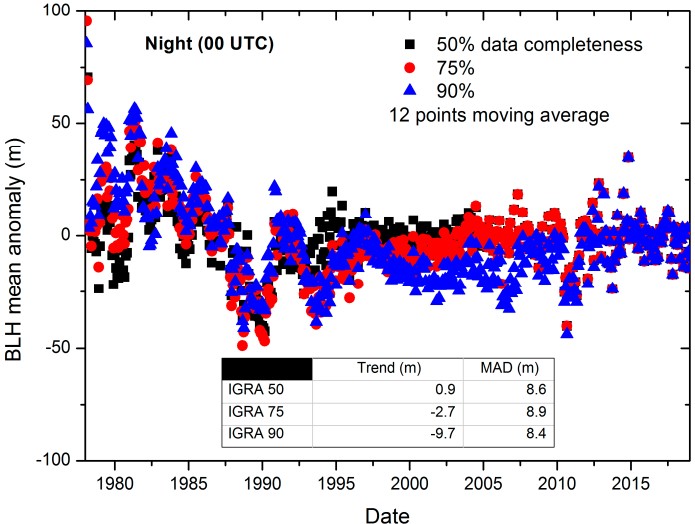

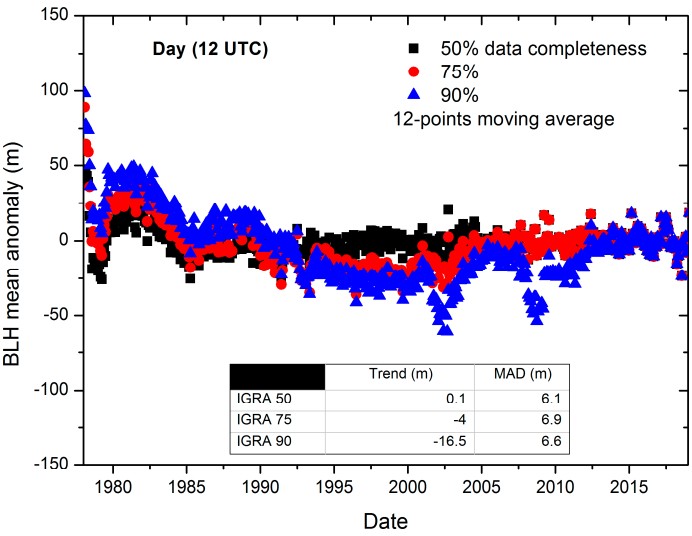

**Figure 6. Top** panel, comparison of the IGRA nighttime BLH mean monthly anomalies for the stations providing time series with a data completeness of 50% (68 stations, black dots), 75% (56 stations, red dots) and 90% (46 stations, blue dots); **bottom** panel, same as **top** panel for daytime anomalies. Both panels embed a table with the estimation of the decadal trend and the median absolute deviation (MAD). All the shown anomalies are obtained applying a 12-point moving average on the original time series.

The comparison reveals a significant impact on the estimation of the trends, quantified to be within about 17 m per decade for the CBL and within 10 m per decade for the NBL if the data completeness changes from 50% to 90%. The latter also implies a change in the number of stations and in the data coverage within the investigated domain. The anomalies show the largest difference in the daytime time series from 1992 to 2004, in between two maxima of a solar cycle with effects in both the Northern and Southern hemispheres [36]. This difference suggests that, for the purpose of estimating BLH trends, the use of time series with a data completeness higher than 75% is recommended. A residual but much smaller effect for the same period is also observed in the nighttime data.

In Table 1, instead, the statistics of the density functions for the values of trends obtained at all the stations for the 50%, 75% and 90% datasets are reported. The median values are not significantly affected by changes in the data completeness, with a difference of only 1.4 m during the day and 0.5 m at night, while the interquartile ranges reveal a skewness in the density function which is more symmetrical for 75% dataset during the day and for 90% dataset at night. Results shown in Figure 6 and Table 1 indicate that uncertainties due to missing data in the time series tend to be systematic, while uncertainties due the spatial completeness/coverage are rather random.

**Table 1.** Median, 1st quartile, 3rd quartile and interquartile (IQ) range for the same dataset shown in Figure 6.

| Day | 1st Quartile | Median da Trend (m) | 3rd Quartile | IQ Range |
|---|---|---|---|---|
| IGRA 50 | −3.1 | −0.5 | 2.7 | 5.8 |
| IGRA 75 | −2.2 | −0.1 | 2.5 | 4.7 |
| IGRA 90 | −1.1 | 0.9 | 2.1 | 3.2 |
| Night | 1st quartile | Median da trend (m) | 3rd quartile | IQ range |
| IGRA 50 | −2.5 | −0.2 | 3.2 | 5.7 |
| IGRA 75 | −2.4 | −0.2 | 3.3 | 5.7 |
| IGRA 90 | −2.8 | −0.7 | 2.6 | 5.4 |

To quantify the sampling uncertainty on BLH trends, due to the limited number of pressure levels in the radiosounding vertical profiles, a comparison between the BLH time series retrieved using the potential temperature gradient method at the Lindenberg station, Germany (WMO index = 10,393, 52.21 N, 14.12 E, 98 m), has been carried out exploiting the availability of both low and high-resolution radiosoundings data at the same station. Indeed, Lindenberg is also a Global Climate Observing System (GCOS) Reference Upper-Air Network (GRUAN) station (www.gruan.org, accessed on 22 February 2021) and, therefore, both IGRA and GRUAN data are available for the comparison. Moreover, Lindenberg has the longest and the most complete time series in GRUAN (processed since 2008). GRUAN data are processed using the GRUAN data processing scheme [37], while IGRA data are processed at each site using the different software of the radiosonde manufacturers. Figure 7 shows the comparison between the ABL heights retrieved from the two datasets in the period 2008–2018. The time series obtained from GRUAN data shows an enhanced variability compared to IGRA, with reduced differences in specific periods, such as before 2009 and from 2014 to 2017. Figure 8 illustrates the two corresponding density functions. The figure shows the effects of this broader variability on the data distribution: both distributions are bimodal with a narrower shape for the IGRA dataset. Nevertheless, the distance between the two peaks of each distribution, which are representative of the maximum probability for the NBL and CBL height, increases from about 200 m to 450 m, when high-resolution data are used, significantly affecting the estimation of the mean and median values for the consider decade. This is also reflected in the estimation of the decadal trend for the two times series which is of −39 m/da for GRUAN (MAD = 200 m) and −115 m/da for IGRA (MAD = 151 m). Although it must be acknowledged that one single station decade of data is not sufficient to properly evaluate a decadal trend—this is also shown by the larger uncertainties affecting the trend estimation (larger MAD values)—

the difference between GRUAN and IGRA demonstrates how parametric uncertainties associated with the lack of vertical resolution of the radiosoundings profiles used to quantify the BLH may strongly influence the magnitude and even the sign of the trend. This links to the results shown in the previous section and the related differences in the methodology used to calculate the BLH. Although it may appear obvious, the use of the highest possible resolution is recommended when BLH estimations obtained from different retrieval methods are compared.

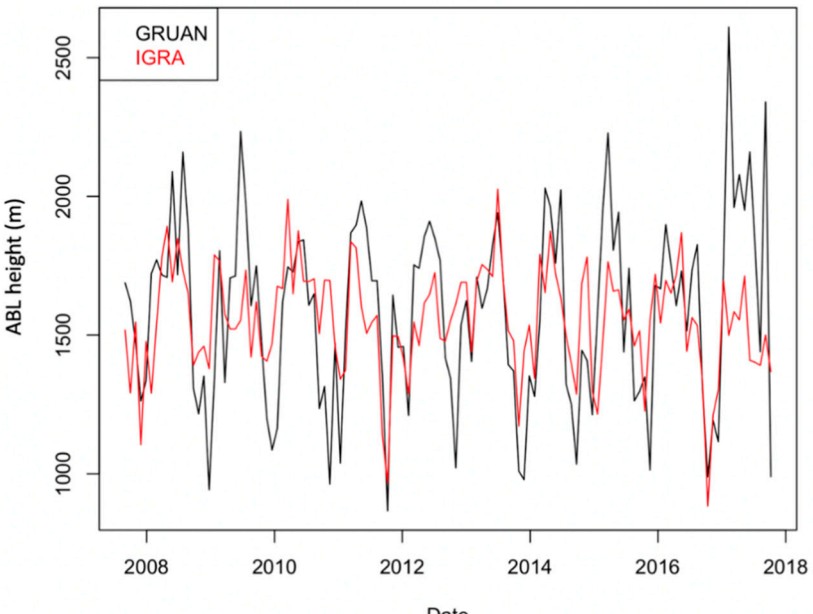

**Figure 7.** Atmospheric boundary layer (ABL) height retrieved using the potential temperature gradient method at the Lindenberg station, Germany (WMO index = 10,393, 52.21 N, 14.12 E, 98 m) for high-resolution radiosounding data provided by the station to the GCOS Reference Upper-Air Network (GRUAN, black line) and for low-resolution data provided to IGRA (red line). Both day and nighttime data are considered. GRUAN temperature profiles are smoothed with an effective vertical resolution [38] of 100 m and this may have a residual effect in the estimation of the BLH.

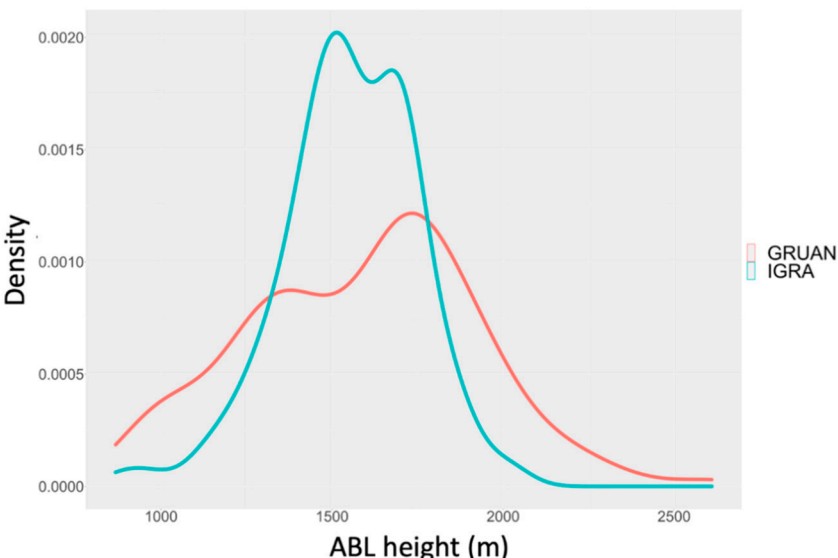

**Figure 8.** Density function of the ABL height estimates shown in Figure 7.

A more accurate quantification of the sampling uncertainties due to the vertical resolution will be provided in future works, considering more stations and more data from the GRUAN database.

## 5. Conclusions

In this paper, the effect due to structural and parametric uncertainties in the estimation of BLH trends obtained from radiosounding profiles in the European Union (EU) mid-latitudes is discussed. In particular, through the use of IGRA and ERA5 atmospheric reanalysis data, two different methods for estimating the BLH height are compared, the former based on the potential temperature gradient method and the latter based on the bulk Richardson number method. The BLH trend estimation are calculated after the removal of the lag-1 autocorrelation for each time series while for ERA5 the autocorrelation is negligible. The lack of a reference dataset [39] and method for estimating the ABL height does not allow us to evaluate which is the most accurate method and dataset in the comparison. The results are summarized in the following points:

(a) Discrepancies between the two BLH retrieval algorithms, used for IGRA and ERA5, are in the order of 800 m during the night and 600 m for the day and are consistent with the results reported in literature.

(b) Comparisons between the BLH trend estimates from IGRA and ERA5 show a good overall consistency with larger discrepancies for the daytime data when ERA5 provides a less homogenous estimation of the CBL in the investigated domain.

(c) The comparison of mean zonal anomalies for all the stations (selected with a data completeness of 75%) shows a difference between the BLH trend estimates for IGRA and ERA which is larger during the day, ranging within −4 m per decade for IGRA and 5 m for ERA5. The anomalies show a larger variability in the 80 s and the 90 s, while variability is lower in the most recent decade. The IGRA dataset shows also a behavior consistent with the solar cycle minimum in the period 1992–2004, which is not observed in ERA5.

(d) Investigation of temporal and spatial sampling uncertainties shows that uncertainties due to the missing data in the time series (i.e., time gaps) tend to be systematic, while uncertainties due the spatial completeness/coverage are random. To minimize sampling uncertainties in the BLH trend analysis based on IGRA data it is recommended to use time series with at least 75% data completeness; below this level of data completeness, uncertainty can increase significantly.

(e) The comparison between the BLH estimates in Lindenberg obtained using the data processed by GRUAN and provided to IGRA, using manufacturer software, demonstrates how parametric uncertainties due to the vertical resolution have a relevant effect on the magnitude and even the sign of the BLH trend. An enhanced quantification of this sampling uncertainty contribution will be provided in forthcoming papers.

The results of this work must be considered as representative of the European mid-latitudes. The extension of the presented results to a sub-domain, such as single countries or smaller regions, may be not immediate due to the representativeness of the stations selected for this study. Moreover, as shown in literature in studies based on climate projections [40], the relationship between forced responses at different spatial scales can be greatly complicated by competition between different forcing factors.

Future investigations based on the use of IGRA data on a global scale are already planned; these may include also comparison with BLH trends estimated from lidar data at a continental scale, ceilometer and GNSS-RO data. Correlations between the trends observed in the BLH and those of other atmospheric variables will also be considered, such as near-surface temperature, relative humidity, and wind speed depending on the availability at different spatial domains in the investigated time period. Aerosol measurements could also be considered to disentangle the climatic signal from the cooling effect of pollution generating the BLH decrease [41].

**Author Contributions:** Conceptualization, F.M. (Fabio Madonna); Data curation, D.S., F.M. (Fabrizio Marra) and Y.W.; Formal analysis, F.M. (Fabio Madonna), D.S. and Y.W.; Methodology, F.M. (Fabio Madonna) and D.S.; Writing–original draft, F.M. (Fabio Madonna); Writing–review & editing, F.M. (Fabio Madonna), P.D.G. and M.R. All authors have read and agreed to the published version of the manuscript.

**Funding:** This research received no external funding.

**Informed Consent Statement:** Not applicable.

**Data Availability Statement:** Data used in this paper are available in a publicly accessible repository: ERA5 data are available in the Climate Data Store (CDS) at https://cds.climate.copernicus.eu, accessed on 22 February 2021, DOI: 10.24381/cds.6860a57; IGRA data are available from the National Centers for Environmental Information of the National Oceanic and Atmospheric Administration at in the https://www1.ncdc.noaa.gov/pub/data/igra/, accessed on 22 February 2021, DOI:10.7289/V5X63K0Q.

**Acknowledgments:** ERA5 data have been obtained from the Copernicus Climate Data Store (CDS). IGRA data are provided by DOC/NOAA/NESDIS/NCDC, National Climatic Data Center, NESDIS, NOAA, U.S. Department of Commerce.

**Conflicts of Interest:** The authors declare no conflict of interest.

# Appendix A

**Table A1.** Station list.

| ID Station | Latitude | Longitude | Height | Country_Code | Country_Name | Start_Time | Stop_Time | Total_Launches_Day | Total_Launches_Night |
|---|---|---|---|---|---|---|---|---|---|
| AGM00060360 | 36.833 | 7.817 | 4 | DZ | Algeria | 02 January 1998 | 16 May 2008 | 0 | 0 |
| AGM00060390 | 36.6833 | 3.2167 | 25 | DZ | Algeria | 01 January 1978 | 12 April 2020 | 0.62 | 0.64 |
| AGM00060419 | 36.283 | 6.617 | 694 | DZ | Algeria | 07 February 1999 | 26 March 2003 | 0 | 0 |
| AGM00060490 | 35.633 | −0.6 | 90 | DZ | Algeria | 18 March 1979 | 14 February 2005 | 0 | 0 |
| AGM00060511 | 35.35 | 1.467 | 989 | DZ | Algeria | 01 October 1987 | 31 December 1988 | 0.6 | 0 |
| AGM00060525 | 34.8 | 5.733 | 87 | DZ | Algeria | 02 January 2000 | 31 December 2002 | 0.21 | 0 |
| AUM00011035 | 48.2486 | 16.3564 | 200 | AT | Austria | 01 January 1978 | 01 October 2019 | 0.98 | 0.98 |
| AUM00011120 | 47.2603 | 11.3439 | 581 | AT | Austria | 23 June 1998 | 01 October 2019 | 0.02 | 0.37 |
| AUM00011240 | 46.9931 | 15.4392 | 340 | AT | Austria | 02 January 1987 | 01 October 2019 | 0 | 0.01 |
| BEM00006400 | 51.083 | 2.65 | 9 | BE | Belgium | 16 July 1999 | 11 September 2002 | 0.01 | 0 |
| BEM00006447 | 50.7969 | 4.3581 | 99 | BE | Belgium | 01 January 1978 | 21 December 2015 | 0.49 | 0.42 |
| BEM00006458 | 50.7456 | 4.7633 | 112.8 | BE | Belgium | 04 January 2005 | 13 April 2020 | 0.06 | 0.91 |
| BEM00006476 | 50.033 | 5.4 | 558 | BE | Belgium | 06 January 1978 | 13 May 2006 | 0.39 | 0.38 |
| BEM00006496 | 50.4822 | 6.1814 | 565.1 | BE | Belgium | 17 February 1999 | 02 December 2009 | 0.03 | 0 |
| BOM00026850 | 53.933 | 27.633 | 231 | BY | Belarus | 01 January 1978 | 30 March 1998 | 0.75 | 0.9 |
| BUM00015614 | 42.65 | 23.3833 | 595 | BG | Bulgaria | 01 January 1978 | 12 April 2020 | 0.92 | 0.45 |
| BUM00015730 | 41.65 | 25.367 | 331 | BG | Bulgaria | 01 January 1978 | 24 April 1991 | 0.75 | 0.61 |
| CYM00017601 | 34.583 | 32.983 | 23 | CY | Cyprus | 02 January 1978 | 12 October 1996 | 0 | 0 |
| CYM00017607 | 35.1408 | 33.3964 | 162 | CY | Cyprus | 01 November 1981 | 27 June 2019 | 0.64 | 0 |
| CYM00017609 | 34.8733 | 33.6172 | 9.8 | CY | Cyprus | 11 November 2003 | 21 June 2006 | 0 | 0 |
| EIM00003953 | 51.9381 | −10.2433 | 23.9 | IE | Ireland | 01 January 1978 | 12 April 2020 | 0.99 | 0.99 |
| EZM00011520 | 50.0078 | 14.4469 | 302 | CZ | Czech Republic | 01 January 1978 | 12 April 2020 | 0.74 | 0.73 |
| EZM00011722 | 49.0833 | 16.6167 | 195 | CZ | Czech Republic | 27 November 1996 | 17 December 2003 | 0.55 | 0.53 |
| EZM00011747 | 49.4525 | 17.1347 | 214.8 | CZ | Czech Republic | 28 November 2003 | 12 April 2020 | 0.92 | 0.92 |
| FRM00007110 | 48.4442 | −4.4119 | 99 | FR | France | 01 January 1978 | 12 April 2020 | 0.83 | 0.75 |
| FRM00007130 | 48.067 | −1.733 | 37 | FR | France | 24 January 1994 | 14 December 1996 | 0.01 | 0.01 |
| FRM00007145 | 48.7744 | 2.0097 | 167 | FR | France | 01 January 1978 | 12 April 2020 | 0.77 | 0.72 |
| FRM00007180 | 48.6833 | 6.2167 | 225 | FR | France | 01 January 1978 | 30 December 2010 | 0.57 | 0.3 |
| FRM00007255 | 47.067 | 2.367 | 161 | FR | France | 01 March 1982 | 01 May 1982 | 1 | 0.95 |
| FRM00007481 | 45.7264 | 5.0778 | 250 | FR | France | 01 January 1978 | 28 February 2011 | 0.55 | 0.52 |
| FRM00007510 | 44.8306 | −0.6914 | 51 | FR | France | 01 January 1978 | 12 April 2020 | 0.71 | 0.69 |
| FRM00007630 | 43.633 | 1.367 | 154 | FR | France | 05 May 1979 | 05 October 1984 | 0.03 | 0.03 |
| FRM00007645 | 43.8569 | 4.4064 | 60 | FR | France | 01 January 1978 | 12 April 2020 | 0.72 | 0.69 |
| FRM00007680 | 43.417 | 6.75 | 6 | FR | France | 01 March 1982 | 01 May 1982 | 0.95 | 0.97 |
| FRM00007690 | 43.65 | 7.2 | 4 | FR | France | 09 September 1999 | 16 November 1999 | 0.46 | 0.46 |
| FRM00007761 | 41.9181 | 8.7928 | 6 | FR | France | 01 January 1978 | 12 April 2020 | 0.73 | 0.69 |
| GIM00008495 | 36.15 | −5.35 | 3 | GI | GIBRALTAR | 01 January 1978 | 02 October 2015 | 0.61 | 0.59 |
| GMM00010046 | 54.3833 | 10.15 | 32 | DE | Germany | 02 January 1978 | 30 September 1994 | 0.31 | 0.21 |
| GMM00010113 | 53.7139 | 7.1525 | 11 | DE | Germany | 20 June 2011 | 12 April 2020 | 0.96 | 0.96 |

**Table A1.** *Cont.*

| ID Station | Latitude | Longitude | Height | Country_Code | Country_Name | Start_Time | Stop_Time | Total_Launches_Day | Total_Launches_Night |
|---|---|---|---|---|---|---|---|---|---|
| GMM00010184 | 54.0978 | 13.4075 | 2 | DE | Germany | 01 January 1978 | 12 April 2020 | 0.71 | 0.7 |
| GMM00010200 | 53.3894 | 7.2269 | 0 | DE | Germany | 02 October 1978 | 31 August 2011 | 0.52 | 0.5 |
| GMM00010238 | 52.8167 | 9.9333 | 70 | DE | Germany | 02 January 1978 | 13 April 2020 | 0.61 | 0.44 |
| GMM00010304 | 52.7333 | 7.3333 | 41.1 | DE | Germany | 02 January 1978 | 20 March 2020 | 0.43 | 0 |
| GMM00010307 | 52.4167 | 7.0667 | 81 | DE | Germany | 02 January 1978 | 29 August 1985 | 0.36 | 0.05 |
| GMM00010338 | 52.45 | 9.7167 | 52 | DE | Germany | 01 January 1978 | 01 June 1997 | 0.58 | 0.57 |
| GMM00010384 | 52.4667 | 13.4 | 50 | DE | Germany | 01 January 1978 | 31 December 1993 | 0.68 | 0 |
| GMM00010393 | 52.2167 | 14.1167 | 112 | DE | Germany | 01 January 1978 | 12 April 2020 | 0.71 | 0.71 |
| GMM00010404 | 51.733 | 6.267 | 43 | DE | Germany | 02 January 1978 | 30 June 1978 | 0.18 | 0.37 |
| GMM00010410 | 51.4056 | 6.9686 | 153 | DE | Germany | 01 January 1978 | 12 April 2020 | 0.7 | 0.7 |
| GMM00010437 | 51.1333 | 9.2833 | 222 | DE | Germany | 02 January 1978 | 02 November 2007 | 0.19 | 0.26 |
| GMM00010468 | 51.55 | 12.0667 | 106 | DE | Germany | 12 September 2000 | 31 August 2006 | 0.68 | 0.71 |
| GMM00010486 | 51.1167 | 13.6833 | 249 | DE | Germany | 01 January 1978 | 11 September 2000 | 0.51 | 0.03 |
| GMM00010548 | 50.5617 | 10.3772 | 450 | DE | Germany | 01 January 1978 | 12 April 2020 | 0.7 | 0.7 |
| GMM00010618 | 49.7 | 7.3333 | 376 | DE | Germany | 02 January 1978 | 13 April 2020 | 0.61 | 0.53 |
| GMM00010659 | 49.75 | 10.2 | 160 | DE | Germany | 01 April 1982 | 05 June 1985 | 0.02 | 0.01 |
| GMM00010687 | 49.7 | 11.95 | 414 | DE | Germany | 05 January 1978 | 01 July 1992 | 0.06 | 0 |
| GMM00010739 | 48.8333 | 9.2 | 314 | DE | Germany | 01 January 1978 | 12 April 2020 | 0.7 | 0.69 |
| GMM00010771 | 49.4283 | 11.9022 | 417 | DE | Germany | 02 January 1978 | 13 April 2020 | 0.61 | 0.48 |
| GMM00010828 | 48.1 | 9.25 | 646 | DE | Germany | 29 March 1995 | 12 October 2007 | 0.05 | 0.19 |
| GMM00010868 | 48.2442 | 11.5525 | 484 | DE | Germany | 01 January 1978 | 12 April 2020 | 0.69 | 0.69 |
| GMM00010921 | 47.9833 | 8.9 | 807 | DE | Germany | 02 January 1978 | 24 June 1994 | 0.31 | 0.33 |
| GMM00010954 | 47.8364 | 10.8722 | 756 | DE | Germany | 28 May 2004 | 20 March 2020 | 0.48 | 0 |
| GMM00010962 | 47.8019 | 11.0119 | 977 | DE | Germany | 01 April 1982 | 10 April 2020 | 0 | 0 |
| GRM00016716 | 37.8897 | 23.7417 | 43.1 | GR | Greece | 01 January 1978 | 12 April 2020 | 0.82 | 0.76 |
| HRM00014430 | 44.0969 | 15.3403 | 78 | HR | Croatia | 01 March 1982 | 12 April 2020 | 0.41 | 0.41 |
| HUM00012812 | 47.267 | 16.633 | 221 | HU | Hungary | 01 March 1982 | 30 April 1982 | 0.8 | 0.75 |
| HUM00012843 | 47.4333 | 19.1833 | 138 | HU | Hungary | 01 January 1978 | 12 April 2020 | 0.65 | 0.69 |
| HUM00012982 | 46.25 | 20.1 | 82 | HU | Hungary | 01 January 1978 | 12 April 2020 | 0.38 | 0.69 |
| ITM00016037 | 46.0303 | 12.5992 | 113 | IT | Italy | 04 January 2010 | 20 December 2011 | 0.2 | 0.5 |
| ITM00016044 | 46.0375 | 13.1883 | 93 | IT | Italy | 01 January 1978 | 16 June 2016 | 0.67 | 0.66 |
| ITM00016045 | 45.9806 | 13.0592 | 50 | IT | Italy | 16 June 2016 | 12 April 2020 | 0.99 | 0.99 |
| ITM00016080 | 45.4614 | 9.2831 | 104 | IT | Italy | 01 January 1978 | 12 April 2020 | 0.69 | 0.7 |
| ITM00016113 | 44.5392 | 7.6125 | 385 | IT | Italy | 28 January 2000 | 12 April 2020 | 0.61 | 0.62 |
| ITM00016144 | 44.6539 | 11.6225 | 10 | IT | Italy | 01 May 1986 | 12 April 2020 | 0.26 | 0.56 |
| ITM00016242 | 41.8 | 12.233 | 3 | IT | Italy | 01 January 1978 | 30 September 1986 | 0.97 | 0.95 |
| ITM00016245 | 41.67 | 12.4508 | 32 | IT | Italy | 01 October 1986 | 12 April 2020 | 0.61 | 0.61 |
| ITM00016320 | 40.6603 | 17.9567 | 14.5 | IT | Italy | 01 January 1978 | 12 April 2020 | 0.68 | 0.69 |

**Table A1.** *Cont.*

| ID Station | Latitude | Longitude | Height | Country_Code | Country_Name | Start_Time | Stop_Time | Total_Launches_Day | Total_Launches_Night |
|---|---|---|---|---|---|---|---|---|---|
| ITM00016429 | 37.9142 | 12.4914 | 7.3 | IT | Italy | 01 January 1978 | 12 April 2020 | 0.67 | 0.61 |
| ITM00016546 | 39.3461 | 8.9675 | 29 | IT | Italy | 28 February 2012 | 12 April 2020 | 1 | 0.99 |
| ITM00016560 | 39.2436 | 9.06 | 4 | IT | Italy | 07 January 1978 | 06 March 2012 | 0.57 | 0.57 |
| LHM00026629 | 54.8839 | 23.8358 | 76.1 | LT | Lithuania | 01 January 1978 | 09 September 2014 | 0.45 | 0.74 |
| LOM00011952 | 49.0333 | 20.3167 | 703 | SK | Slovakia | 01 January 1978 | 12 April 2020 | 0.77 | 0.77 |
| MDM00033815 | 47.017 | 28.867 | 170 | MD | Republic of Moldova | 01 January 1978 | 11 October 2002 | 0.58 | 0.64 |
| MKM00013586 | 41.95 | 21.633 | 233 | MK | The former Yugoslav Republic of Macedonia | 08 January 1982 | 27 September 2008 | 0.01 | 0.16 |
| MTM00016597 | 35.85 | 14.483 | 91 | MT | Malta | 01 January 1978 | 31 December 1978 | 1 | 0.99 |
| NLM00006210 | 52.167 | 4.433 | 1 | NL | Netherlands | 10 May 1994 | 20 November 2002 | 0.05 | 0 |
| NLM00006235 | 52.9269 | 4.7811 | 1.2 | NL | Netherlands | 13 September 2000 | 11 October 2014 | 0 | 0 |
| NLM00006260 | 52.0989 | 5.1797 | 1.9 | NL | Netherlands | 01 January 1978 | 12 April 2020 | 0.75 | 0.9 |
| PLM00012120 | 54.7536 | 17.5347 | 1.8 | PL | Poland | 01 January 1978 | 12 April 2020 | 0.88 | 0.93 |
| PLM00012330 | 52.417 | 16.85 | 84 | PL | Poland | 01 January 1978 | 27 March 1992 | 0.9 | 0.92 |
| PLM00012374 | 52.4078 | 20.9564 | 94.2 | PL | Poland | 01 January 1978 | 12 April 2020 | 0.98 | 0.98 |
| PLM00012425 | 51.1131 | 16.8811 | 119.6 | PL | Poland | 01 January 1978 | 12 April 2020 | 0.66 | 0.95 |
| POM00008579 | 38.7667 | −9.1333 | 104 | PT | Portugal | 01 January 1978 | 01 January 2020 | 0.77 | 0.28 |
| RIM00013275 | 44.7667 | 20.4167 | 203 | RS | Serbia | 01 January 1978 | 12 April 2020 | 0.56 | 0.81 |
| RIM00013388 | 43.3333 | 21.9 | 203 | RS | Serbia | 01 April 2016 | 12 April 2020 | 0.99 | 0.99 |
| ROM00015120 | 46.7778 | 23.5714 | 410 | RO | Romania | 01 January 1978 | 24 October 2012 | 0.34 | 0.82 |
| ROM00015420 | 44.5106 | 26.0781 | 90 | RO | Romania | 01 January 1978 | 12 April 2020 | 0.73 | 0.92 |
| UKM00003414 | 52.8 | −2.667 | 76 | GB | United Kingdom of Great Britain | 08 October 1997 | 25 December 2000 | 0 | 0.22 |
| UKM00003496 | 52.6833 | 1.6833 | 13 | GB | United Kingdom of Great Britain | 01 January 1978 | 19 March 2001 | 0.54 | 0.53 |
| UKM00003501 | 52.4167 | −4 | 92 | GB | United Kingdom of Great Britain | 19 February 2001 | 19 February 2001 | 0 | 0 |
| UKM00003502 | 52.1394 | −4.5711 | 133 | GB | United Kingdom of Great Britain | 03 January 1978 | 23 March 2020 | 0.12 | 0.01 |
| UKM00003559 | 52.1039 | −0.4214 | 29 | GB | United Kingdom of Great Britain | 09 October 1996 | 07 April 2020 | 0 | 0 |
| UKM00003590 | 52.117 | 0.967 | 87 | GB | United Kingdom of Great Britain | 25 January 1999 | 21 March 2006 | 0.01 | 0.08 |
| UKM00003649 | 51.75 | −1.583 | 88 | GB | United Kingdom of Great Britain | 14 July 1992 | 30 July 2005 | 0 | 0.08 |
| UKM00003693 | 51.5547 | 0.8269 | 2 | GB | United Kingdom of Great Britain | 03 January 1978 | 20 December 2010 | 0.12 | 0 |
| UKM00003715 | 51.4 | −3.35 | 67 | GB | United Kingdom of Great Britain | 01 December 1989 | 20 December 1997 | 0 | 0.05 |

**Table A1.** *Cont.*

| ID Station | Latitude | Longitude | Height | Country_Code | Country_Name | Start_Time | Stop_Time | Total_Launches_Day | Total_Launches_Night |
|---|---|---|---|---|---|---|---|---|---|
| ROM00015480 | 44.217 | 28.633 | 13 | RO | Romania | 01 January 1978 | 08 October 2001 | 0.49 | 0.76 |
| RSM00026702 | 54.7264 | 20.5583 | 19 | RU | Russian Federation | 01 January 1978 | 12 April 2020 | 0.51 | 0.62 |
| RSM00026781 | 54.75 | 32.0667 | 240 | RU | Russian Federation | 01 January 1978 | 12 April 2020 | 0.8 | 0.92 |
| SIM00014015 | 46.0656 | 14.5122 | 299 | SI | Slovenia | 08 March 1996 | 12 April 2020 | 0 | 0.01 |
| SPM00008001 | 43.3658 | −8.4214 | 58 | ES | Spain | 01 January 1978 | 12 April 2020 | 0.67 | 0.67 |
| SPM00008023 | 43.4911 | −3.8006 | 52 | ES | Spain | 01 October 1986 | 12 April 2020 | 0.62 | 0.57 |
| SPM00008160 | 41.6786 | −1.0731 | 252 | ES | Spain | 09 April 1982 | 31 October 2015 | 0.54 | 0.53 |
| SPM00008190 | 41.3844 | 2.1181 | 95 | ES | Spain | 14 November 2007 | 12 April 2020 | 0.95 | 0.95 |
| SPM00008221 | 40.4653 | −3.5797 | 631 | ES | Spain | 01 January 1978 | 12 April 2020 | 0.7 | 0.7 |
| SPM00008301 | 39.55 | 2.617 | 6 | ES | Spain | 06 July 1988 | 17 December 2002 | 0.62 | 0.61 |
| SPM00008302 | 39.6058 | 2.7067 | 41 | ES | Spain | 03 January 1978 | 12 April 2020 | 0.56 | 0.56 |
| SPM00008430 | 38.0019 | −1.1708 | 61 | ES | Spain | 01 August 1984 | 12 April 2020 | 0.65 | 0.59 |
| SZM00006610 | 46.8117 | 6.9425 | 490 | CH | Switzerland | 01 January 1978 | 27 August 2019 | 0.73 | 0.72 |
| TSM00060715 | 36.8333 | 10.2333 | 4 | TN | Tunisia | 01 January 1978 | 12 April 2020 | 0.43 | 0.43 |
| TSM00060750 | 34.717 | 10.683 | 21 | TN | Tunisia | 23 March 1998 | 23 March 1998 | 0 | 0 |
| TUM00017064 | 40.9 | 29.15 | 18 | TR | Turkey | 01 January 1978 | 12 April 2020 | 0.93 | 0.95 |
| TUM00017130 | 39.95 | 32.8833 | 891 | TR | Turkey | 01 January 1978 | 11 April 2020 | 0.95 | 0.96 |
| TUM00017220 | 38.4333 | 27.1667 | 25 | TR | Turkey | 01 January 1978 | 12 April 2020 | 0.86 | 0.9 |
| TUM00017240 | 37.75 | 30.55 | 997 | TR | Turkey | 01 January 1978 | 12 April 2020 | 0.88 | 0.89 |
| UKM00003257 | 54.3 | −1.533 | 40 | GB | United Kingdom of Great Britain | 01 May 1990 | 24 December 2000 | 0.01 | 0.02 |
| UKM00003322 | 53.55 | −2.9167 | 56 | GB | United Kingdom of Great Britain | 01 January 1978 | 31 March 1996 | 0.61 | 0.6 |
| UKM00003354 | 53.0056 | −1.2511 | 117 | GB | United Kingdom of Great Britain | 23 July 1998 | 12 April 2020 | 0.55 | 0.84 |
| UKM00003377 | 53.167 | −0.517 | 70 | GB | United Kingdom of Great Britain | 03 July 1989 | 04 July 2005 | 0.01 | 0.02 |
| UKM00003743 | 51.2017 | −1.8058 | 132 | GB | United Kingdom of Great Britain | 03 January 1978 | 18 March 2020 | 0.36 | 0 |
| UKM00003774 | 51.0833 | −0.2167 | 144 | GB | United Kingdom of Great Britain | 01 January 1978 | 30 September 1992 | 0.76 | 0.75 |
| UKM00003808 | 50.2183 | −5.3275 | 87 | GB | United Kingdom of Great Britain | 01 January 1978 | 12 April 2020 | 0.71 | 0.7 |
| UKM00003882 | 50.8994 | 0.3169 | 52 | GB | United Kingdom of Great Britain | 01 February 1993 | 12 April 2020 | 0.39 | 0.66 |
| UKM00003918 | 54.5 | −6.3333 | 18 | GB | United Kingdom of Great Britain | 25 June 2002 | 11 April 2020 | 0.53 | 0.89 |
| UKM00003920 | 54.4833 | −6.1 | 37 | GB | United Kingdom of Great Britain | 01 January 1978 | 24 June 2002 | 0.57 | 0.55 |

**Table A1.** *Cont.*

| ID Station | Latitude | Longitude | Height | Country_Code | Country_Name | Start_Time | Stop_Time | Total_Launches_Day | Total_Launches_Night |
|---|---|---|---|---|---|---|---|---|---|
| UKM00033966 | 45.0464 | 34.5989 | 204.6 | GB | United Kingdom of Great Britain | 01 January 2011 | 30 December 2013 | 0 | 0.45 |
| UPM00033317 | 50.1667 | 27.0333 | 277 | UA | Ukraine | 01 January 1978 | 12 April 2020 | 0.31 | 0.64 |
| UPM00033345 | 50.4 | 30.5667 | 166 | UA | Ukraine | 01 January 1978 | 12 April 2020 | 0.86 | 0.95 |
| UPM00033393 | 49.8167 | 23.95 | 319 | UA | Ukraine | 01 January 1978 | 12 April 2020 | 0.36 | 0.63 |
| UPM00033631 | 48.633 | 22.267 | 118 | UA | Ukraine | 01 January 1978 | 31 December 2008 | 0.48 | 0.77 |
| UPM00033791 | 48.0333 | 33.2167 | 123 | UA | Ukraine | 01 January 1978 | 12 April 2020 | 0.75 | 0.32 |
| UPM00033837 | 46.4333 | 30.7667 | 42 | UA | Ukraine | 01 January 1978 | 12 April 2020 | 0.33 | 0.73 |
| UPM00033946 | 44.6833 | 34.1333 | 180 | UA | Ukraine | 01 January 1978 | 04 November 2010 | 0.45 | 0.77 |

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
