# Peer review of "Assessment of Trends and Uncertainties in the Atmospheric Boundary Layer Height Estimated Using Radiosounding Observations over Europe"

_atmosphere, doi:10.3390/atmos12030301_

Round 1
Reviewer 1 Report
Please, see the attached review.

Author Response
In attached pdf file, a point-by-point reply to the reviewer's comments is provided.

Reviewer 2 Report
This study analysed the uncertainties in the atmospheric2boundary layer height over Europe. IGRA and ERA5 data were used to analyse the atmospheric2boundary layer height. The study performed a comprehensive analysis and the manuscript is technically sound for publication in the atmosphere. The originality of the manuscript is sufficient for publication. However, some minor issues need to be incorporated before it’s final publication
- The first sentence of the abstract could have been attractive and simple sentence. Authors may re-write it with a little background. Authors should avoid complex sentence.
- Abstract is well written and contains many informations. Authors should include some key findings in the abstract section.
- Line 129, authors should define IGRA first and then use abbreviation.
- Line 145, for standard reference pressure author, should include a reference.
- Figure 3, authors need to discuss a bit more. Why is the IRGA density profile significantly different for day and night time?
Author Response

(The authors gave the same response as above.)
